# A Study on the Layout of Hospital Ward Buildings in Cold Regions of China Based on the Efficiency of Nurse Rounds

Qingtan Deng [1], Chenxia Jiao [1,*], Guangbin Wang [1], Xiaoyi Song [1] and Jiayao Zang [2]

[1] School of Architecture and Urban Planning, Shandong Jianzhu University, Jinan 250101, China; dengqingtan@126.com (Q.D.)
[2] School of Civil Engineering and Architecture, Qingdao Agricultural University, Qingdao 266041, China
* Correspondence: 2021055213@stu.sdjzu.edu.cn

**Abstract:** As an important public facility, the number, area, and scale of hospital buildings are growing rapidly. The efficiency of nurses' rounds to beds is an important indicator of the efficiency of nursing units in ward buildings. Ward buildings occupy a very important position in the overall energy consumption of hospital building complexes. The type and scale of nursing unit floorplans are some of the key factors affecting the energy consumption of ward buildings. In this paper, three typical floorplan layout types of hospital ward buildings in cold regions of China are selected. The relationships between rounding efficiency, building energy consumption, floorplan layout, and building size were quantified using Origin based on linear regression and non-linear regression. The study showed that at 60 beds, the efficiency of nurse rounds was 35.68% higher in the double-corridor layout compared to the single-corridor. At 44 beds, the difference in average bed energy consumption between the double-corridor type and the single-corridor type is the greatest, with a 9.02% saving in energy consumption. This result confirms that the layout and scale of the ward building has a significant impact on the efficiency of nursing unit rounds and building energy efficiency.

**Keywords:** ward buildings; nursing units; cold regions; layout types; rounding efficiency; building energy consumption; regression analysis

## 1. Introduction

As a special public building to meet the basic needs of the nation, hospitals have a complex functional composition, diversified spatial types, high density of pedestrian flow, and unstable personnel, among which the ward building has the largest floor area scale, accounting for 37~41% of the total floor area of a general hospital [1]. With the increasing prominence of the aging population in China, the number of patients requiring hospital admission has increased, which has led to a continuous surge in the number of hospital beds and a rapid increase in floor space (Figure 1), while directly affecting the size of nursing units in ward buildings. Due to the impact of COVID-19, more people are opting for online consultations and home treatment, resulting in a decrease in hospital admissions in 2020 and 2021 compared to 2019 [2]. This increase in the size of the nursing units temporarily relieves the space requirements of inpatients but may cause inconvenience to the medical and nursing staff. This raises the question of how to balance the increasing size of the ward building with the efficiency of staff rounds.

As the most basic treatment unit in a hospital and the space that patients spend the most time in, the quality of the design of the nursing unit will have a direct impact on the efficiency of care provided by the medical and nursing staff and therefore on the patient's treatment and recovery process. Therefore, a key objective of nursing unit design should be to minimize nurse travel and increase patient contact to improve the efficiency of care. Academic research on nursing units in ward buildings has mainly focused on the nursing unit design [3,4], the planar pattern of nursing units [5,6], and the spatial environment

of hospital nursing units [7]. The above studies are mostly qualitative and empirical in nature. Most of the studies on nurse rounding routes are based on field observation [8,9], and simulation experiments are rarely used. In this paper, Revit was selected to simulate the closest distance from the center of the nurse's station to each bed by creating a planar model of the nursing unit and thus evaluating the efficiency of the rounds quantitatively.

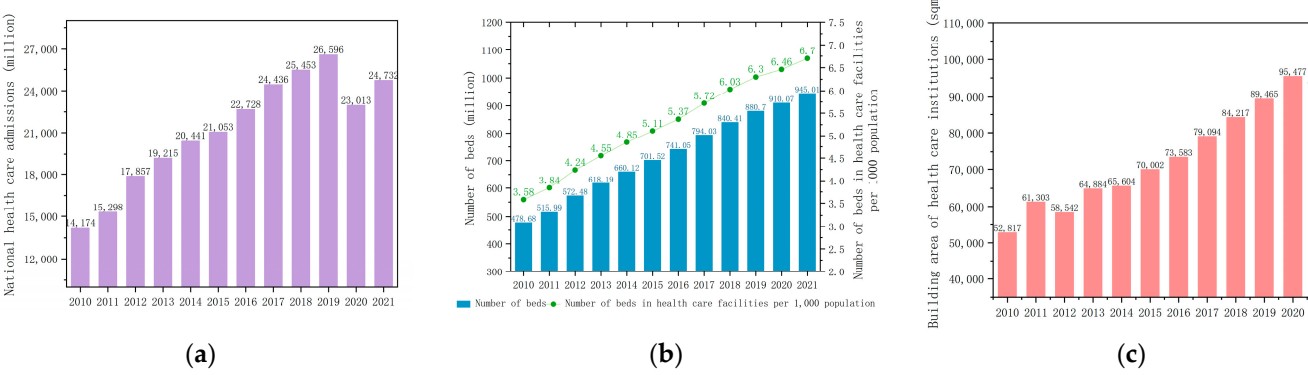

**Figure 1.** Changes in hospital admissions, number of beds, and floor space in the last decade. (compiled and drawn from the China Health Care Statistics Yearbook 2012–2022). (**a**) National healthcare admissions, (**b**) Number of beds and Number of beds in health care facilities per 1000 population, (**c**) Building areas of healthcare institutions.

The different floorplan types and building sizes (number of beds) of nursing units affect not only the walking behavior and distance of nurses but also the building's consumption of energy. There is a relative lack of research on the impact of the type of floorplan layout and building size of nursing units in hospital inpatient buildings on nurse rounding efficiency and building energy consumption. This study aims to investigate the relationship between the efficiency of nursing rounds and building energy consumption in hospital ward buildings and the type and scale of nursing unit plan layout through a combination of qualitative and quantitative approaches. On this basis, the study aims to propose the best solution for different situations to achieve low carbon energy efficiency.

The study selects the cold regions of China as the study area and establishes a floorplan model for nursing units based on the common types of floorplans in cold regions (single-corridor, double-corridor, and radial layout). In this paper, we take the design of floor space layout in hospital scheme design as the starting point. In the first part of the paper, the ward buildings in the cold regions of China were categorized and typical floorplan types were extracted, and the floorplan model was established by controlling the number of beds. In the second part, we use Revit to analyze the nurse rounding paths of the established floorplan model, and calculate three control indicators for assessing rounding efficiency; quantification of building energy consumption in ward buildings using Design Builder. The third part, based on the simulation results, further analyses the impact of different layout types and building sizes of nursing units on the efficiency of nurses' rounds and building energy consumption, puts forward suggestions for the layout of ward buildings in cold regions, and finally gives the significance of the study and conclusion.

## 2. Research Object

### 2.1. Study Area

"Thermal Design Code for Civil Buildings" divides China's building climate zoning into mild regions, hot summer and warm winter regions, cold regions, and severe cold regions (Figure 2). The range of cold regions mainly includes Tianjin, Tianjin, Hebei, Ningxia, Shanxi, Shandong, most of Shaanxi, central-eastern Gansu, southern Liaoning, southern Xinjiang, Anhui, Henan, southern Tibet, and northern Jiangsu [10]. In China, cold regions have an average temperature of −10–0 °C in January and 18–28 °C in July; regions where the average annual daily temperature exceeds 25 °C for less than 80 days

and the average annual daily temperature is below 5 °C for between 90 and 145 days. The energy consumption in cold regions accounts for about 30% of the national building energy consumption and is among the more energy-intensive regions in the five climate zones, and it is maintaining a growing trend [11]. Therefore, it is even more necessary to control building energy consumption in cold regions. Jinan is one of the most representative cities in the colder regions of China, so the city of Jinan (36.40° N, 117.00° E) was chosen as the study site.

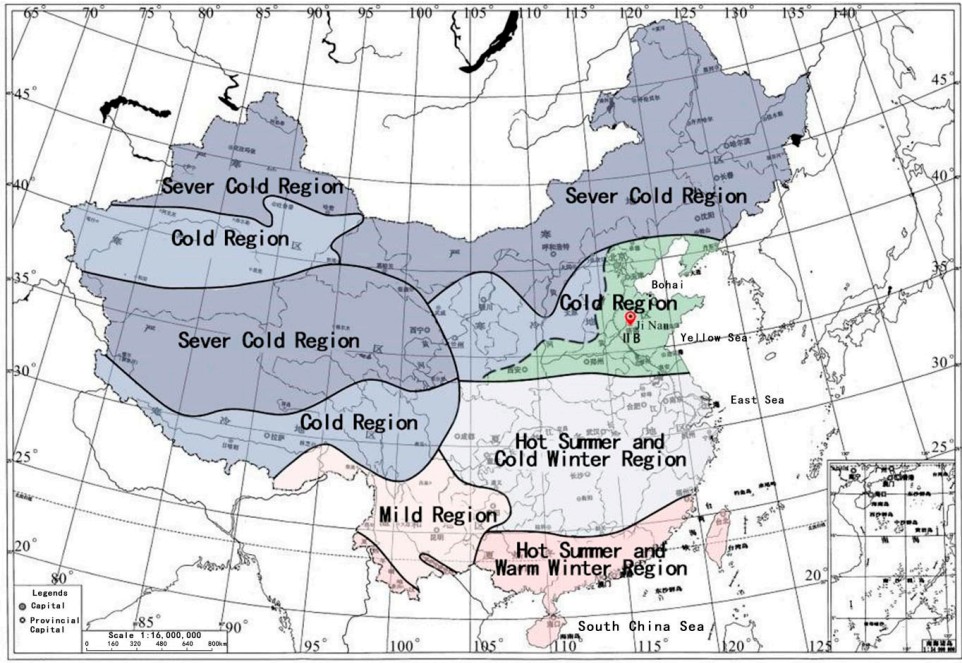

**Figure 2.** Thermal design zoning of buildings in China.

### 2.2. Research Object Classification

Ward buildings are generally designed as high-rise, with little change in shape concavity and convexity, little change in window-to-wall ratio on the facade, high similarity in internal space, and little change in layout type, and their nursing units are mainly located on the fourth floor and above, which are the basic units of ward buildings, consisting of a set of fully equipped personnel (doctors, nurses, workers), several patient beds, related diagnostic and treatment facilities, as well as ancillary medical, living, management, and transportation rooms, with independence of use [12]. In this paper, nursing units are divided into four main spaces based on their functional use: ward space, nurses' station space, support space, and public transport space (Figure 3).

The layout of the nursing unit was first used in an open layout, but later developments were represented by the Nightingale Ward at St Thomas' Hospital in London, UK, which gradually changed from an open layout to the standard layout that is more common today. Nursing units are divided into general nursing units, intensive care and leukemia units, burn units, psychiatric units, infectious disease units, and obstetrics and pediatric units, depending on their target groups [12], etc. Since general nursing units account for a larger proportion of the units and the other units, except for general nursing units, have a higher degree of functional complexity. This paper focuses on general ward nursing units, the size of which is measured by the number of beds.

The plan type of nursing unit evolved from the initial single-corridor plan to single-corridor, double-corridor, and radial layout to reduce the rounding distance. Triangular and circular plan layouts were adopted to further improve the efficiency of medical staff, reduce the distance between nurses' stations and wards, and increase the number of nursing

patients, but this type of plan layout type has a poor sense of direction and a large shape factor, which is not conducive to building energy efficiency [13].

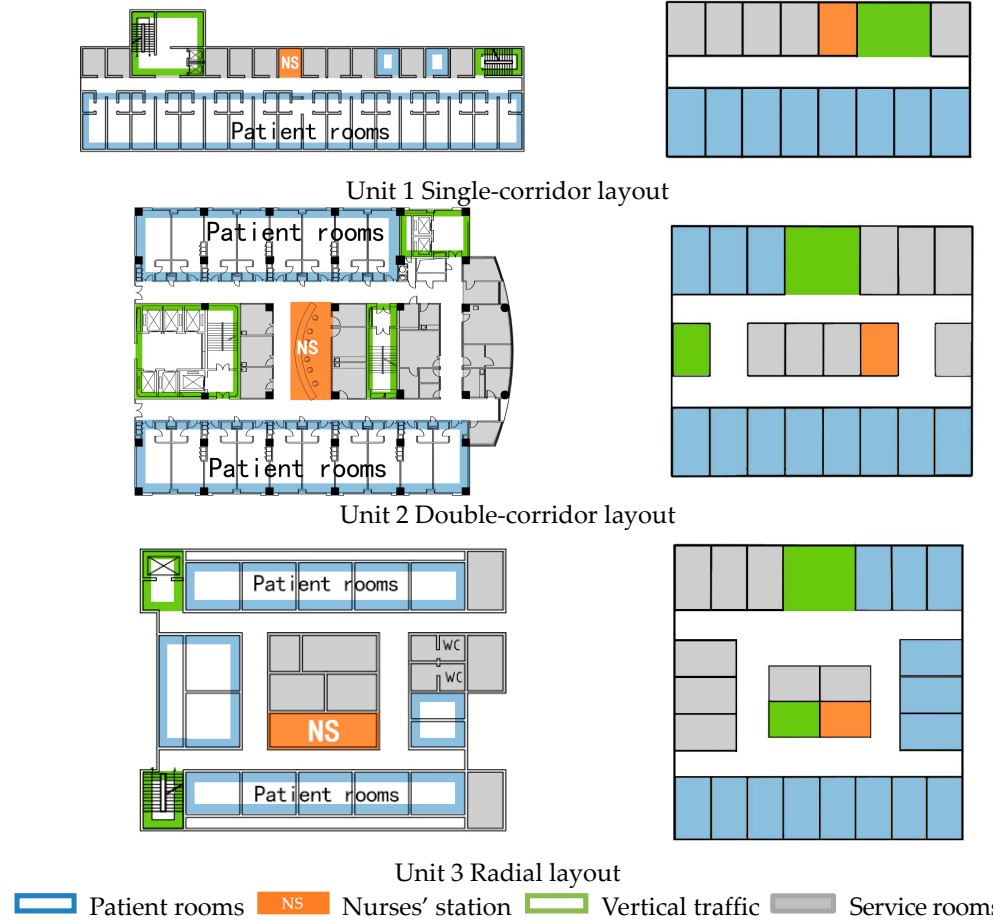

Unit 1 Single-corridor layout

Unit 2 Double-corridor layout

Unit 3 Radial layout

Patient rooms    NS Nurses' station    Vertical traffic    Service rooms

**Figure 3.** Three types of floorplan analysis: single-corridor, double-corridor, and radial layout.

This paper selects general hospitals in typical cities in cold regions of China as a sample, and based on the research results the types of inpatient nursing unit floorplans can be summarized as single corridor, double corridor, radial layout, circular, and polygonal units, etc. Compared with samples from other climatic regions, there are fewer types of inpatient nursing unit layouts in cold regions, with single-corridor, double-corridor, and radial layout types accounting for a larger number of samples. Therefore, in this paper, the three most common types of rectangular floorplans, single-corridor, double-corridor, and radial layout, are selected for analysis and study (Figure 3).

(1)    Single-corridor layout

The single-corridor layout is used to connect the various functional rooms with a single corridor. The stairwell is generally located at both ends of the building, with the nurses' station in a moderate position, and the rooms on both sides are provided with good natural light and ventilation. It has been shown that indoor air quality, good natural ventilation, and the quality of natural light affect the health of patients [14,15]. The orientation is clearer in the interior and there is a high degree of traffic recognition. However, with larger nursing units or an increase in the number of different functions, the traffic space in the interior is lengthened and with it the difference in distance between the nurses' station and the furthest and nearest ward increases.

(2)    Double-corridor layout

The double-corridor layout is composed of two corridors linking the functional rooms, with a compact layout. The middle room and vertical traffic can serve both sides. The wards

are arranged on the north and south sides, breaking through the long, monotonous form of aisles, shortening the distance of care, and improving the efficiency of care. According to statistics, the single-corridor nursing unit is about 1.24 times as long as the double-corridor unit, while the double-corridor unit has 65% more corridor length than the single-corridor unit [12]. However, the rooms in the middle cannot be naturally ventilated and it is difficult to form a natural ventilation system throughout the nursing unit, thus requiring mechanical ventilation and artificial lighting, which increases energy consumption.

(3)　Radial layout

The radial layout nursing unit is a departure from the traditional long building form, based on the double corridor, and is further lengthened and widened so that the traffic space is no longer a purely linear space with one way to the end. The floorplan is more compact, with wards arranged around the building and ancillary areas and vertical traffic arranged in the center of the building, placing the nurses' station in the middle section, achieving the shortest walking distance for nursing care while also providing the nursing station with the best viewpoint for observation and monitoring [16], increasing the visibility of the nurses' station to patients and allowing direct observation of patients, thus reducing the walking distance required by nurses [17]. However, it is difficult to identify directions inside the building, which can easily lead to disorientation, and there are more east-west facing wards.

## 3. Methodology

This paper simulates the effect of different nursing unit floorplans and the number of beds on rounding efficiency and body shape factor and then uses statistical analysis to derive the relationship between different floorplans and sizes and rounding efficiency and building energy consumption (Figure 4).

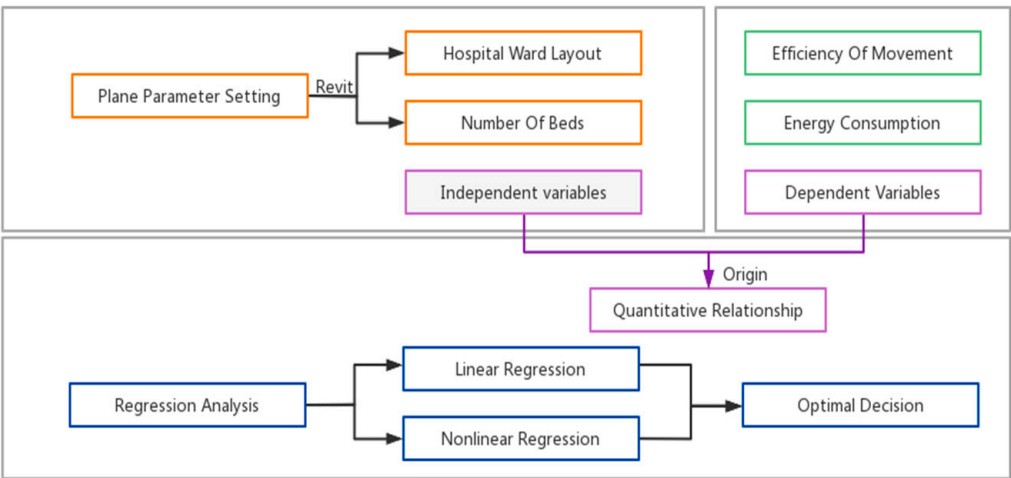

**Figure 4.** Planar parameter setting, software simulation, and result analysis process.

This study uses regression analysis to analyze the effect of changes in nursing unit size and floorplan on the efficiency of rounding and building energy consumption. First, a linear regression analysis was conducted on the dependent variables (rounding efficiency, building energy consumption) and the independent variables (floorplan, number of beds), respectively. Linear regression is an analytical method that uses linear regression equations to model the relationship between one or more independent and dependent variables [18] and measures the strength of the statistical relationship between the variables through the Pearson correlation coefficient. The Pearson correlation coefficient $\rho$ is a linear correlation coefficient, and the value of $\rho$ ranges from $-1$ to 1. When $\rho > 0$, it indicates a positive correlation between the two variables, i.e., the two variables change in the same direction; when $\rho < 0$, it indicates a negative correlation between the two variables, i.e., the

two variables change in opposite directions. The closer ρ is to 1, the higher the correlation between the two variables, and the closer it is to 0, the lower the correlation between the two variables [19]. Second, for dependent and independent variables with poor linear correlation, curve estimation was performed using SPSS to determine and derive the best-fitting and significant data model by F-value and *p*-values to find the relationship between the independent and dependent variables. The regression analysis requires *p*-values of 0.05 or less, with a highly significant correlation if the *p*-values are less than 0.01. Additionally, the closer the F-value of the correlation is to 1, the smaller the difference, and the larger F-value the larger the difference. For the F-value of the regression analysis, a larger value means that the regression analysis is statistically significant [20]. According to the results of the regression analysis obtained, it can be concluded that there is a relationship between the independent variables and the dependent variable, and the best floorplan and building size for rounding efficiency and building energy consumption is selected in terms of both functional space and building performance.

### 3.1. Modeling of Different Plan Types of Nursing Units

The object selected for this paper is a hospital ward building in a cold region, and the first three types of floorplan layout nursing units are modeled. The whole model building process is as follows: first, the number of beds is determined. Second, the total floor area of the hospital is determined according to the requirements of the average floor area per bed index. Thirdly, the total area of the ward building is calculated according to the proportion of the inpatient department to the total floor area (Table 1). Then the area of the nursing unit is determined according to the number of building floors. The length and width are further determined according to the different floor types, and then the floorplan design is carried out.

**Table 1.** Indicators and ratios of average floor space per bed in general hospitals (drawn according to the "Construction Standards for General Hospitals" 2021).

| Bed Size | 200 Beds to 499 Beds | 500 Beds to 799 Beds | Sector | Inpatient Department |
|---|---|---|---|---|
| Floor space per bed indicator($m^2$/bed) | 113 | 116 | Proportion of average bed floor area target (%) | 37–41 |

To facilitate the calculation and modeling, the inpatient department is set up on 10 floors with 1 nursing unit on each floor and a floor height of 4.2 m [21,22]. There are 10 nursing units in total. The efficiency simulation model for this experiment was incremented from 30 to 60 beds in steps of 2, for a total of 16 sets of simulation models. From this, it was possible to determine the specific building design requirements (Table 2).

According to the findings, the activity parameters of the hospital's ward areas are more demanding than those of other functional areas. To avoid large discrepancies between the simulated and actual energy consumption of the building due to inconsistent ward areas and to avoid excessively long care routes, the calculated floor area per floor needs to be adjusted. The area share of nursing unit wards was set at 0.4 based on the statistics [6], so the calculated total floor area per floor was multiplied by 0.4 to obtain the floor area of the ward area, ensuring consistency between the area of the ward and the number of wards.

The formulae for calculating the total area of the ward block, the floor area per floor, and the area of individual nursing units are as follows:

$$S = N_1 \cdot S_b \tag{1}$$

S—Hospital floor space ($m^2$)
$N_1$—Number of beds (bed)
$S_b$—Average floor space per bed ($m^2$/bed)

$$S_t = S \cdot P \tag{2}$$

$S_t$—Total area of ward block (m$^2$)
S—Hospital floor space (m$^2$)
P—Proportion of ward block to average bed floor space target

$$S_f = S_t / N_2 \tag{3}$$

$S_f$—Area per ward floor (m$^2$)
$S_t$—Total area of ward block (m$^2$)
$N_2$—Number of floors in the building

$$N_3 = S_f \cdot F / S_i \tag{4}$$

$N_3$—Number of wards per nursing unit
$S_f$—Area per ward floor (m$^2$)
F—Ward area share factor
$S_i$—Area of individual ward (m$^2$)

**Table 2.** 16 parameter settings for the group model.

| | Number of Beds in the Nursing Unit | Total Number of Beds | Area per Floor (m$^2$) | Number of Wards in the Nursing Unit |
|---|---|---|---|---|
| **1** | 30 | 300 | 970 | 11 |
| **2** | 32 | 320 | 1035 | 12 |
| **3** | 34 | 340 | 1099 | 13 |
| **4** | 36 | 360 | 1164 | 14 |
| **5** | 38 | 380 | 1229 | 15 |
| **6** | 40 | 400 | 1293 | 16 |
| **7** | 42 | 420 | 1398 | 17 |
| **8** | 44 | 440 | 1472 | 18 |
| **9** | 46 | 460 | 1539 | 19 |
| **10** | 48 | 480 | 1615 | 20 |
| **11** | 50 | 500 | 1686 | 21 |
| **12** | 52 | 520 | 1800 | 22 |
| **13** | 54 | 540 | 1870 | 23 |
| **14** | 56 | 560 | 1939 | 24 |
| **15** | 58 | 580 | 2008 | 25 |
| **16** | 60 | 600 | 2077 | 26 |

*3.2. Three Evaluation Indicators of Rounding Efficiency*

The nursing work of health care staff usually consists of two types of work: first, looking after patients, observing their condition, and carrying out general nursing treatment. Second, making routine rounds and caring for patients' activities such as living, eating, and sleeping. To meet these requirements, the nursing unit should provide as convenient and quick a route for care as possible. Long rounding routes have been one of the most critical factors affecting nurse fatigue and stress in the nursing process [23,24]. The literature suggests that nurses who spend more time walking spend less time at the patient's bedside [25–27]. The nurses walked the most between the nurses' station and the ward [28,29]. In an observational study, the time spent on rounds between the nurses' station and the wards accounted for 34.5% of the total time spent by nurses on 12-h shifts [23]. A nurse walks an average of 2.4 to 3.4 miles per day during the day shift [25], and up to 6 miles [30]. Some researchers believe that less walking time would contribute to more patient care activities and better patient outcomes [31].

In the early 1970s, the American Medical Planning Association (MPA) and Bobrow/Thomas Associates (BTA) adopted a combination of control indicators for nursing

unit design distances. These made the center of the nurses' most common and much-needed daily work activities the core of care in the nursing unit. The main control indicators are the average distance from the nursing activity center to the ward, the distance factor, and the distance difference from the nursing activity center to the furthest and nearest ward, which are used to evaluate the efficiency of the nursing unit rounds [12].

(1) The average distance from the nursing activity center to the bed determines the combined efficiency of the different floorplan types and the formula.

$$D_{avg} = D_{sum}/n \tag{5}$$

$D_{avg}$—Average distance from care activity center to bed (m); $D_{sum}$—Sum of the distance from the nursing activity center to each bed (m); n—Number of beds

(2) The distance from the nursing activity center to the furthest bed and the nearest bed is an indicator designed to promote a more balanced distance from the nursing activity center to the bed.

$$D_{diff} = D_f - D_n \tag{6}$$

$D_{diff}$—Distance difference (m); $D_f$—Distance from the Care Nursing Center to the furthest ward (m); $D_n$—Distance from the Care Nursing Center to the nearest ward (m)

(3) The average distance coefficient from the center of nursing activity to the bed is used to compare efficiency at different numbers of beds and reflects the relationship between nursing efficiency and hospital size, the indicator is expressed as follows.

$$D_c = D_{avg}/n \tag{7}$$

$D_c$—Distance coefficient (m); $D_{avg}$—Average distance from care activity center to bed (m); n—Number of beds

If the average distance is shorter and the distance coefficient is smaller, the more efficient the nursing round is; if the difference in distance from the nurses' station to the beds is smaller, it means that the distance from the nurses' station to the wards is more balanced and therefore the more efficient the nursing round is, and vice versa. These three indicators suggest that the key factor in improving nursing unit efficiency is to reduce the distance between nurses' stations and wards to increase the number of patients cared for and the number of beds, thus achieving the highest efficiency of rounds. Most hospital designs to date still use shortened routes of care as an important control for measuring the efficiency of nursing units.

*3.3. Building Energy Consumption Simulation Software and Selection of Energy Consumption Evaluation Indexes for Ward Buildings*

The energy consumption simulations were carried out using Design Builder, a comprehensive building energy use simulation tool based on the Energy Plus algorithm. It can perform multi-performance simulations on the physical models created and can predict and evaluate the performance of a solution at any stage of the design process. Sun et al. used Design Builder v6 to simulate the physical environment and energy consumption of a model of a hospital outpatient department with a courtyard space in a cold region, to explore the main design elements of the courtyard space in a hospital outpatient department, the influence mechanism of building performance and the optimization design ideas [32]. Rahman uses Design Builder to simulate the energy consumption of an air conditioning system in an Australian hospital building and analyses the energy savings and economics of the air conditioning system using various energy-saving measures [33]. Shi used Design Builder v6 to simulate the quantitative relationship between building layout and energy consumption in a tertiary hospital in a cold region, and verified the energy consumption from overall to local, from functional zoning to departmental layout, thus verifying the reliability of the simulation tool [9].

Energy consumption per unit of floor space is the most commonly used indicator of energy efficiency, and a large number of scholars have conducted statistics and calculated this indicator for different types of hospitals in different regions. Energy consumption in some hospitals in cold regions ranges from 92.4 to 627.41 kWh/(m$^2$·a) [34–36]. Du et al. analyzed the correlation between energy consumption and various indicators, and concluded that in terms of the reliability of the indicators, the indicator of the number of beds per unit > the indicator of hospitalization per unit > the indicator of floor space per unit > the indicator of outpatient visits per unit [37]. Considering that bed count statistics are more convenient and reliable, this paper uses the average bed energy consumption index to evaluate hospital energy consumption, and its calculation formula is as follows:

$$E_b = E_t/n \qquad (8)$$

$E_b$—Energy consumption per bed [kWh/(hb·a)]
$E_t$—Total Energy consumption [kWh/a]
$N$—Number of beds [hb]

## 4. Simulation Process and Analysis of Results

### 4.1. Relationship between Type and Size of Nursing Unit Layout and Efficiency of Rounds

Traditionally, in nursing unit studies, walking distances have been estimated based on point-to-point linear proximity between key areas [8,30]. Among Korean inpatient unit case studies, Shin and Kang (2016) assessed nurse walking distances using traditional point-to-point linear measurements based on field interviews of nurses' patient room assignments [9]. However, human walking behavior typically does not follow the centerline and makes a curve at corners to move along the shortest path to a destination point [38]. With the application and popularity of computer-aided tools in architectural design, visual simulation software has been developed to find the shortest path and path optimization. Song et al. used Revit 2020 to develop a crowd emergency evacuation path optimization model, which significantly improved the efficiency of evacuation optimization [39]. Minguk Kim et al. used machine learning to simulate the pathfinding methods of real construction workers and eventually obtained the workers' path of travel [40]. The Autodesk Revit platform has developed Path of Travel, which can model the path of travel based on the Revit 2020 platform and can generate the shortest path trajectory and travel time for a pedestrian between two selected points. Therefore, for this study, the path of travel was selected to simulate the shortest path that best reflects the true behavioral pattern of the nurse from the central point of the nurse's station to each patient's bed.

According to the overall design requirements of the ward building determined in Section 3.1, three types of nursing units, namely single-corridor, double-corridor, and radial layout, are used for the layout. The location of the wards needs to facilitate effective access from the nurses' station and other ancillary rooms [29]. Therefore, in the layout, the nurses' station is placed in the moderate part of the nursing unit, while the rooms with a higher number of beds are placed closer to the nurses' station, to minimize the nurses' rounds [41]. Using the Path of Travel point in Revit, the nurse's path of travel was created by taking the center of the nurses' station as the starting point and the patient bed as the endpoint. Revit automatically identifies obstacles such as nurses' tables and walls as well as ward doors, generates the shortest paths, and can count the distance of all paths, resulting in 48 plan sketches (Table 3).

The lengths of all the paths plotted were counted using the Schedule tool in Revit. Finally, the data obtained were used to calculate three measures of rounding efficiency using statistical software: the average distance between the nurses' station and the beds, the distance difference, and the distance coefficient (Table 4).

Based on the statistical results of the three indicators of touring efficiency, the number of beds as the independent variable and the average distance, distance difference, and distance coefficient as the dependent variables were analyzed. The relationship between the three indicators of rounding efficiency and the number of beds for different floor types

was derived, and a one-dimensional linear regression equation was obtained, respectively, to discover the law of action between the type of floor space and the number of beds and rounding efficiency, to guide the design of nursing units in ward buildings. This will guide the design of ward-building nursing units.

**Table 3.** Visualization of shortest paths for the three floorplan types simulated.

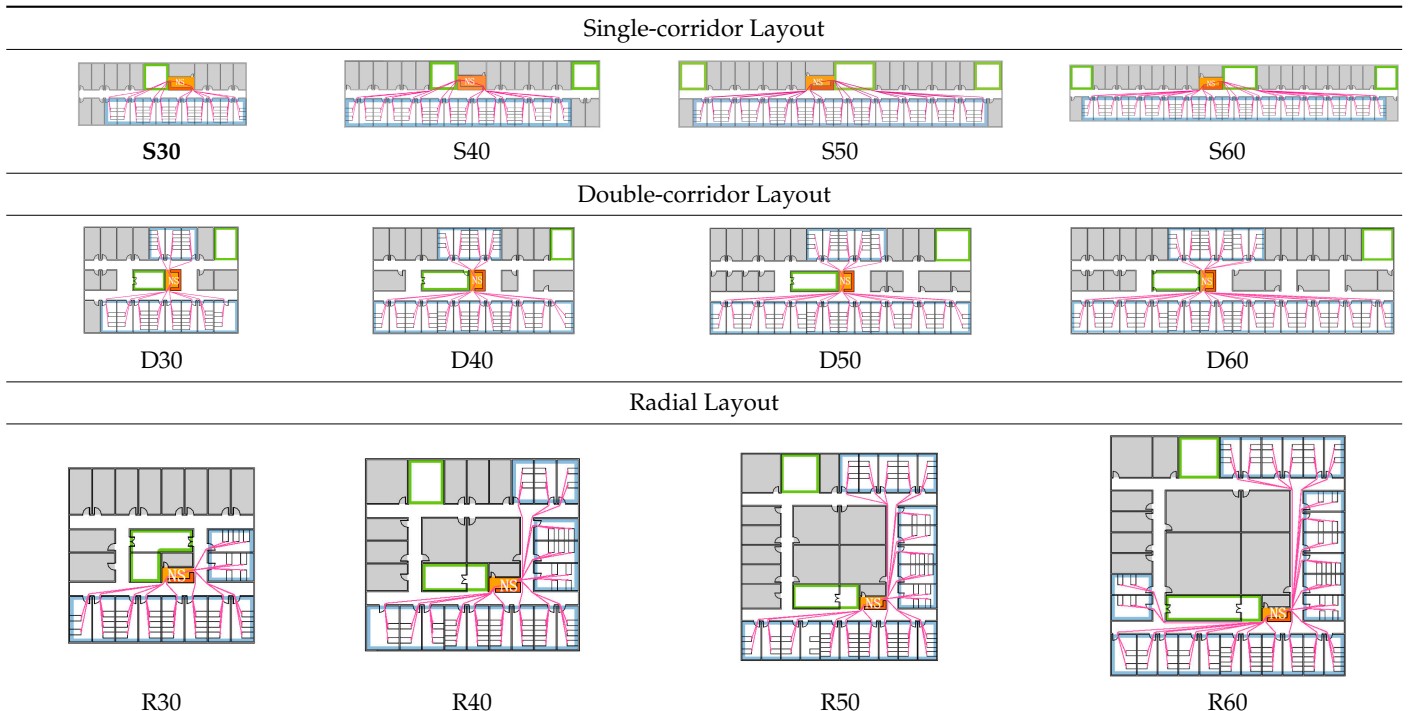

**Table 4.** Average distance and distance factor statistics for the three floorplan types.

| | Single-corridor Layout | | | Double-corridor Layout | | | Radial Layout | | |
|---|---|---|---|---|---|---|---|---|---|
| Number of Beds | Average Distance (m) | Distance Coefficient (m) | Distance Difference (m) | Average Distance (m) | Distance Coefficient (m) | Distance Difference (m) | Average Distance (m) | Distance Coefficient (m) | Distance Difference (m) |
| 30 | 15.76 | 0.5214 | 25.21 | 14.23 | 0.4743 | 19.13 | 15.67 | 0.5223 | 23.42 |
| 32 | 16.41 | 0.5000 | 25.20 | 14.39 | 0.4497 | 19.14 | 16.71 | 0.5222 | 28.27 |
| 34 | 17.30 | 0.5095 | 27.31 | 14.65 | 0.4309 | 19.13 | 17.65 | 0.5191 | 29.44 |
| 36 | 17.84 | 0.5003 | 27.15 | 14.61 | 0.4058 | 18.98 | 18.53 | 0.5147 | 32.41 |
| 38 | 19.01 | 0.5056 | 31.35 | 15.78 | 0.4153 | 23.60 | 19.29 | 0.5132 | 34.43 |
| 40 | 19.86 | 0.4965 | 34.95 | 16.13 | 0.4033 | 23.59 | 20.14 | 0.5133 | 37.39 |
| 42 | 20.55 | 0.4946 | 35.02 | 16.18 | 0.3852 | 23.55 | 21.51 | 0.5121 | 42.94 |
| 44 | 21.39 | 0.4893 | 34.99 | 16.84 | 0.3827 | 26.99 | 22.50 | 0.5114 | 44.41 |
| 46 | 22.48 | 0.4965 | 37.71 | 17.02 | 0.3700 | 27.01 | 23.57 | 0.5124 | 47.36 |
| 48 | 23.33 | 0.4861 | 39.27 | 18.13 | 0.3777 | 31.67 | 24.50 | 0.5104 | 50.40 |
| 50 | 24.31 | 0.5002 | 45.25 | 17.89 | 0.3578 | 31.54 | 25.99 | 0.5106 | 56.42 |
| 52 | 25.31 | 0.4887 | 47.24 | 18.21 | 0.3502 | 31.63 | 26.58 | 0.5112 | 58.48 |
| 54 | 27.77 | 0.5340 | 53.02 | 18.65 | 0.3454 | 31.71 | 28.52 | 0.5485 | 59.53 |
| 56 | 29.53 | 0.5679 | 55.02 | 18.83 | 0.3363 | 31.80 | 30.1 | 0.5788 | 61.72 |
| 58 | 31.15 | 0.5990 | 54.96 | 19.36 | 0.3338 | 31.81 | 32.31 | 0.6214 | 66.47 |
| 60 | 33.52 | 0.6446 | 57.04 | 19.98 | 0.3330 | 31.86 | 35.16 | 0.6762 | 68.58 |

(1) Comparison of nurse rounding efficiency by floorplan type for the same number of beds

Based on the above statistics, it can be seen from the graph that there is a correlation between the efficiency of nursing unit rounds and the floorplan layout pattern, and the number of beds. For the same number of beds, the average distance from the nurses' station to the beds is further in the single-corridor type compared to the other two layout types, indicating that this type of layout pattern is less efficient in terms of nursing care, while the double layout and radial layout are more efficient in terms of rounding (Figure 5a).

In Figure 5b, the balance of the nursing unit layout is reflected by the calculation of the difference between the distance from the nurses' station to the furthest and nearest ward, where the double-corridor and the radial layout have a smaller difference in distance for the same number of beds, indicating that their nurses' stations have a better balance of distance to each ward.

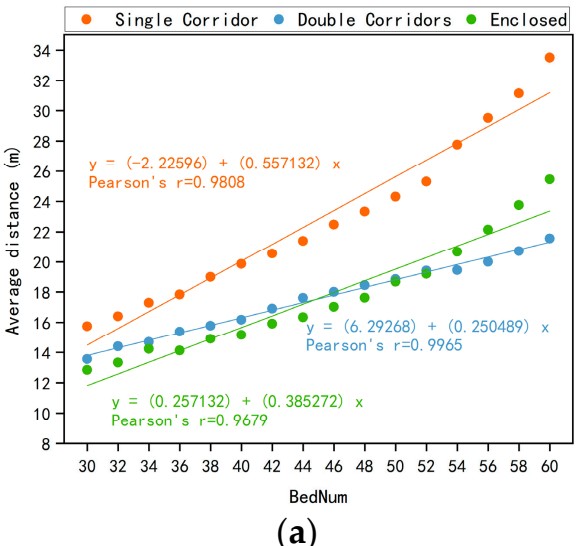

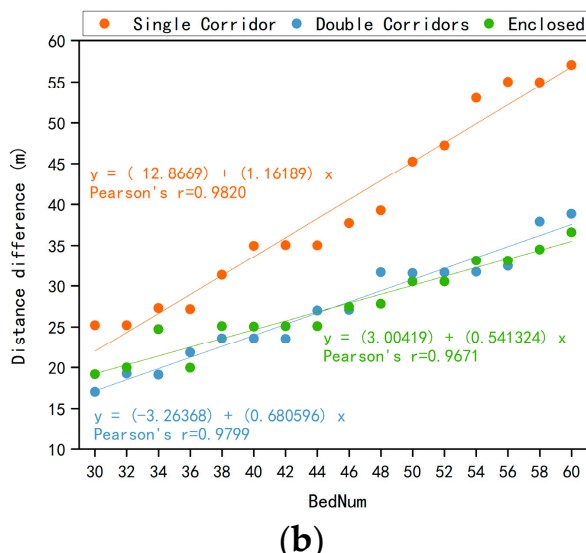

**(a)**  **(b)**

**Figure 5.** Three flat forms of nurse rounding efficiency: (**a**) Average distance; (**b**) Distance difference.

The results of the study show that: as can be seen in Figure 5, there is a significant linear relationship between the number of beds and the average distance and distance difference of the three floorplan types, and the Pearson correlation coefficient is close to 1, indicating a high correlation. (1) In the case of the same building size (i.e., the same number of beds), when the number of beds is less than 45, the three floorplan types are ranked from high to low in terms of the efficiency of nurse rounds as radial layout > double-corridor type > single-corridor. When the number of beds is greater than 45, the three floorplan types are ranked from highest to lowest in terms of rounding efficiency: double-corridor layout > radial layout > single-corridor layout. (2) When the number of beds is the same, the three floorplan types are ranked from highest to lowest in terms of balance: double-corridor layout, radial layout > single-corridor layout. The greatest increase in rounding efficiency was seen in the double-corridor floorplan (19.98) compared to the single corridor (33.52) at 60 beds, with a 40.39% increase in rounding efficiency.

(2) Comparison of the efficiency of rounds with different numbers of beds in the same floorplan type

According to the conditions for statistical use of the multiple linear regression equation, first, the data obtained for the distance coefficients were entered into the mathematical statistical analysis software SPSS 26 to create scatter plots of the number of beds versus distance coefficients for different floor types. Next, Linear and Quadratic curve estimation was performed to obtain the fitted plots and model summary (Table 5) of distance coefficients versus the number of beds for single-corridor type, double-corridor type, and radial layout type, respectively.

Table 5 gives a summary of the models and it can be seen that the best-fit $R^2$ is for the quadratic term model (0.892/0.987/0.894). The quadratic term models for the three floorplan distance coefficients are more significant than the primary term models in terms of F-values and *p*-values. It is intuitive from the scatter plots in the table that the quadratic term model fits the original observations the best of the two curves fitted by the curvilinear model. Therefore, the quadratic term model is the most appropriate data model for the experiment.

**Table 5.** Summary of distance coefficient fit plots and model summary.

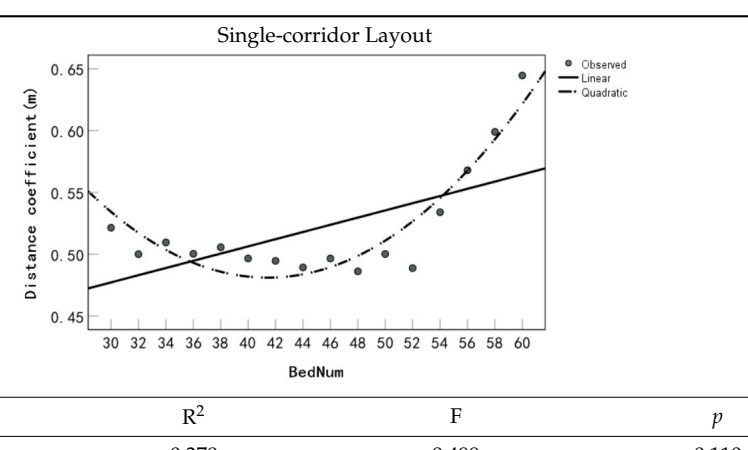

|  | $R^2$ | F | *p* |
|---|---|---|---|
| Linear | 0.378 | 8.499 | 0.110 |
| Quadratic | 0.875 | 45.484 | 0.000 |

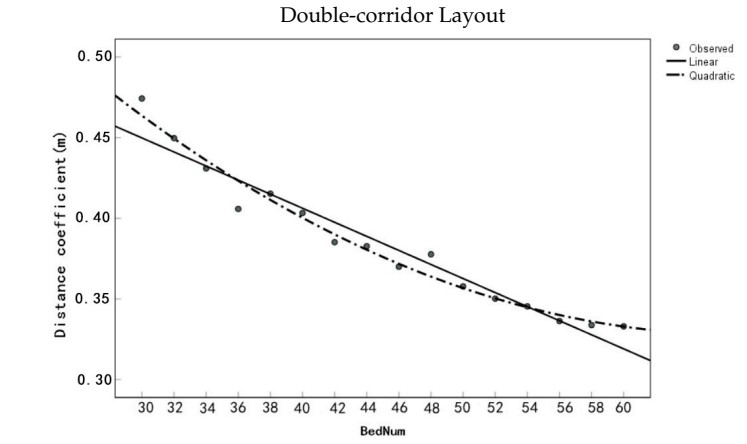

|  | $R^2$ | F | *p* |
|---|---|---|---|
| Linear | 0.942 | 225.857 | 0.000 |
| Quadratic | 0.974 | 244.073 | 0.000 |

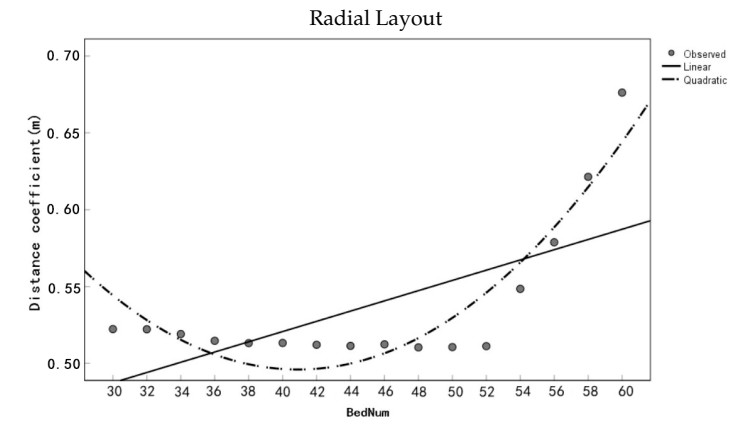

|  | $R^2$ | F | *p* |
|---|---|---|---|
| Linear | 0.434 | 10.745 | 0.005 |
| Quadratic | 0.868 | 42.813 | 0.000 |

The fitted distance coefficient curve regression equation indicates that, for a given layout type of floorplan, the efficiency of rounds increases and then decreases as the number of beds increases (Table 5). The regression coefficient for the single-corridor layout has a

smaller value compared to the double-corridor layout and the radial layout, indicating that increasing or decreasing the same number of beds has a greater impact on the efficiency of rounding in the single-corridor layout and, to a lesser extent, in the radial layout. The distance between the number of newly added beds and the nurses' station gradually increases as the location of the nearest bed remains the same, making the distance difference gradually increase, indicating that the balance decreases as the number of beds increases (Figure 5b). In summary, the efficiency of rounding on the same floorplan type requires a combination of distance factors and distance differences.

### 4.2. Relationship between the Layout and Size of Nursing Unit Layout and Building Energy Consumption

#### 4.2.1. Energy Consumption Simulation Parameter Settings

The simulation process and the analysis of the results were carried out in four steps: the establishment of a simulation model for the energy consumption of the ward building; the setting of parameters for the simulation of the energy consumption of the ward building; the simulation of the energy consumption of the building; and the statistical analysis of the correlation between the variables.

This section examines the impact of the layout and size of the nursing unit on the energy consumption of the building, so other factors that affect the energy consumption of the building need to be set consistently. According to the results of the survey, it is found that the orientation of ward buildings in cold areas is mostly south, so this paper selects a window-to-wall ratio of 0.3 for uniform setting, and the rest of the envelope parameters and indoor activity parameters of nursing units are uniformly set as shown in Table 6 [11]. The 48 models created in 3.1 were simulated using Design Builder v6 and the relevant parameters were set to simulate energy consumption throughout the year.

**Table 6.** Setting of thermal performance parameters and indoor activity parameters for inpatient units in cold regions.

| Thermal Performance Parameters of the Envelope | | Indoor Activity Parameters | | | | | | |
|---|---|---|---|---|---|---|---|---|
| Structure | Heat Transfer Coefficient W/(m²·K) | Regional Functions | Room Setting Temperature in Summer | Set Temperature for Winter Rooms | Setting Humidity in Summer Rooms | Set Humidity in Winter Rooms | Staff Density (Person/m²) | Lighting Power Density |
| Roofing | 0.4 | Auxiliary rooms | 27 (°C) | 18 (°C) | 65% | 30% | 0.125 | 9 (W/m²) |
| Ground | 0.45 | Ward | 27 (°C) | 21 (°C) | 65% | 30% | 0.17 | 5 (W/m²) |
| Exterior walls | 0.45 | Corridor | 27 (°C) | 18 (°C) | 60% | 35 | 0.10 | 5 (W/m²) |
| External windows | 2.4 | Nurse's station | 26 (°C) | 20 (°C) | 60% | 40 | 0.13 | 9 (W/m²) |

#### 4.2.2. Building Energy Simulation Results

This section uses SPSS 26 software to perform curve estimation on the energy consumption simulation results of 48 building models for three floorplan types in ward buildings. First, the obtained data on average bed energy consumption were input into SPSS to build scatter plots of average bed energy consumption versus the number of beds for different planar forms, and curve estimation of linear, quadratic, and cubic was performed on them to obtain fitted plots and model summary of average bed energy consumption versus the number of beds for the single-corridor, double-corridor, and radial layout, respectively (Table 7).

A summary of the model simulations is given in Table 7 and it can be seen that the best-fit $R^2$ is for the cubic term model (0.983/0.967/0.970). In terms of F-values and *p*-values, the three-term models for the three flat energy consumptions were more significant than the primary and secondary term model fits. From the scatter plots in the table, it is quite intuitive that of the two curves fitted by the curve model, the curve fitted by the cubic term model fits the original observations the best. Therefore, the cubic term model is the most appropriate data model for the experiment.

**Table 7.** Summary of average energy consumption per bed fit plots and model summary.

### Single-corridor Layout

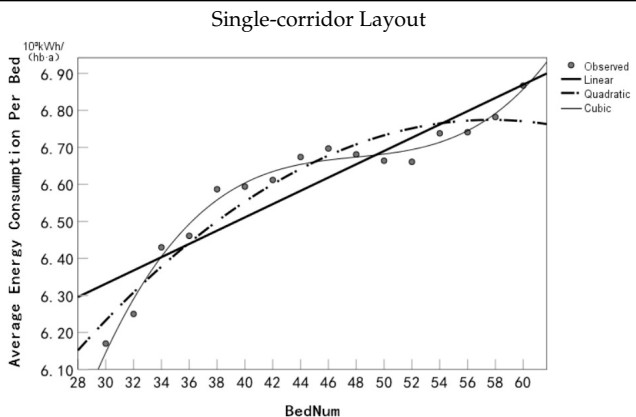

|  | $R^2$ | F | *p* |
|---|---|---|---|
| Linear | 0.827 | 66.952 | 0.000 |
| Quadratic | 0.913 | 68.627 | 0.000 |
| Cubic | 0.983 |  | 0.000 |

### Double-corridor Layout

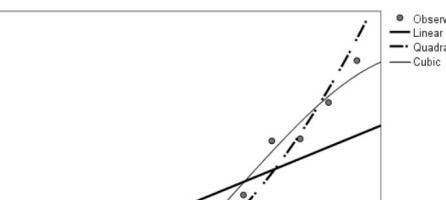

| Linear | $R^2$ | F | *p* |
|---|---|---|---|
| Quadratic | 0.463 | 12.069 | 0.004 |
| Cubic | 0.902 | 59.870 | 0.000 |
| Linear | 0.967 |  | 0.000 |

### Radial Layout

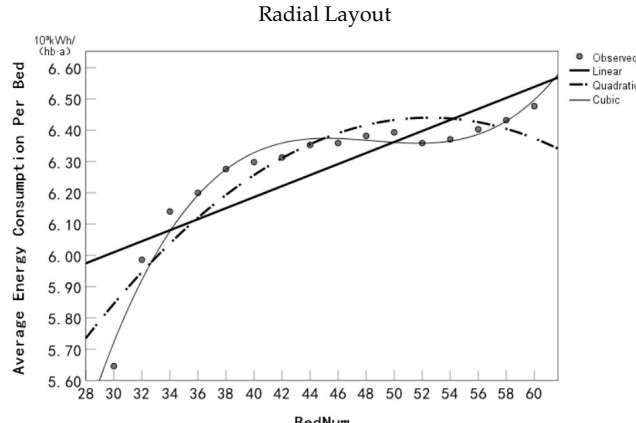

|  | $R^2$ | F | *p* |
|---|---|---|---|
| Linear | 0.658 | 26.952 | 0.300 |
| Quadratic | 0.855 | 38.227 | 0.000 |
| Cubic | 0.970 |  | 0.000 |

Both single-corridor and radial layouts tend to increase energy consumption per bed as the number of beds increases. The double-corridor type has a clear trend of decreasing and then increasing average bed energy consumption as the number of beds increases, with the lowest average bed energy consumption at 38 beds. As can be seen in Figure 6, the greatest energy savings are achieved when the number of beds is 44, with a 9.02% energy saving using the double gallery type compared to the single gallery type.

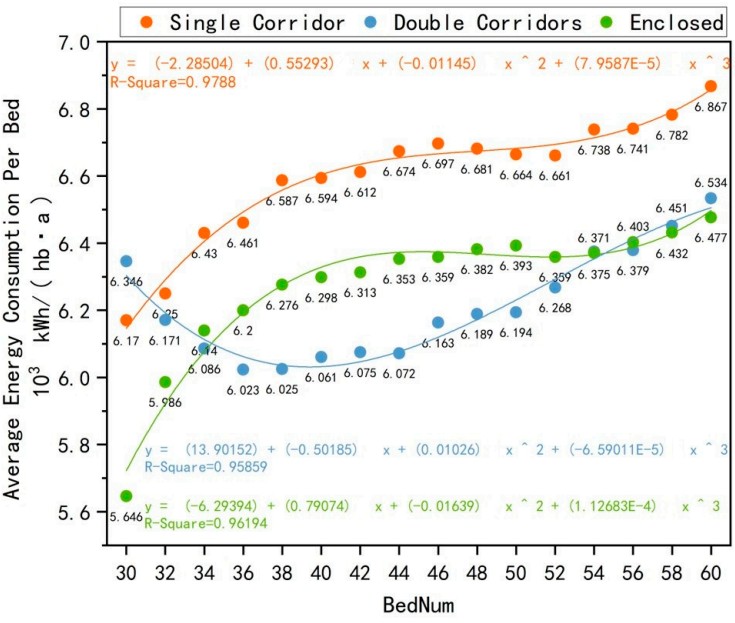

**Figure 6.** Energy consumption for all three floorplan types of bed.

In addition, studies have shown that carbon emissions during the operational phase of a building can be analyzed based on building energy consumption and carbon emission factors for power generation [42]. According to a study on building energy consumption in China, the average carbon emission factor of the power grid in North China (Jinan belongs to North China) is 0.8843 kg/(kW·h) [43]. The formula for building carbon emissions is:

$$EM = F \cdot E \tag{9}$$

EM—Carbon emissions from buildings (kg)
E—Building energy consumption (kW·h)
F—Regional grid carbon emission factors kg/(kW·h)

Therefore, when the number of beds is 44, the difference in average bed energy consumption between the double and single gallery type is 602 kW·h/(hb), and the total building energy consumption is 26,488 kW·h lower, which is equivalent to a reduction in carbon emissions of 23,423 kg, which can be calculated by the above building carbon emissions formula. This shows that it is important to choose the best form of layout to reduce the building's energy consumption and reduce its carbon footprint.

## 5. Discussions

### 5.1. Efficiency of Nurse Rounds

(1)     Same number of beds, preferred type of floorplan

When designing the building, to ensure the efficiency of the rounds, when the number of beds is less than 45, the priority is to choose the radial layout, followed by the double-corridor type, the single-corridor type has the lowest efficiency of the rounds. When the number of beds is greater than 45, the double corridor is preferred, followed by the radial layout, with the single corridor still being the least efficient for cruising. Single-corridor rounds are the least efficient because, for the same number of beds, single-corridor wards

are mostly arranged in one direction, significantly increasing the flow of nursing care for nurses.

(2) Same floorplan, preferred range of number of beds

When the layout form of the floorplan has been determined, if the single-corridor type is chosen, the number of beds is best in the range of 40–44 rounding efficiency, the double-corridor type gives preference to beds in the range of 56–60, and the radial layout has the best number of beds in the range of 42–60. However, as the number of beds increases, the balance of its nursing units decreases, as the distance between the increased number of beds and the nurses' station gradually increases, while the location of the nearest beds remains the same, making the distance difference gradually increase. The nurse's station should be positioned in the middle of the nursing unit so that more wards can be observed, and rooms with a high number of beds should be placed as close as possible to the nurse's station to improve the efficiency of the rounds. Separate nurses' stations should be provided when the distance from the nurses' station to the farthest ward is greater than 30 m [13].

*5.2. Building Energy Consumption*

(1) When the number of beds is 30–34 and greater than 54, the radial layout is preferred; when the number of beds is 34–54, the double-corridor type is preferred. In the case of the same building scale, the single-corridor type has a larger volume factor compared with the double-corridor type and the radial layout, and therefore its building energy consumption is larger.

(2) Both the single-corridor type and the radial layout tend to increase with the increase in the number of beds. The double-corridor type has a clear tendency to decrease and then increase as the number of beds increases, and the lowest energy consumption per bed is at 40 beds.

*5.3. The Best Floorplan Should Be Chosen by Combining Rounding Efficiency and Building Energy Consumption*

In summary, the number of beds in the 30~34 and 54~60 radial layouts has the best roving efficiency and the lowest average energy consumption per bed. The number of beds is between 34 and 45, and the difference between the double-corridor and the radial layout is small in terms of the efficiency of rounds, while the energy consumption of the building is significantly lower in the double corridor than in the ring corridor. In addition, the largest difference is that at 46 beds, the double-corridor type is 129 kWh/(hb) more energy efficient than the radial layout, so the double-corridor type should be preferred at 34~45. The number of beds in the 45~54 double gallery type has the best rounding efficiency and the lowest energy consumption per bed. The single gallery type is not recommended as a preferred floorplan type due to its lower rounding efficiency and higher average bed energy consumption compared to the other two floorplan forms.

**6. Conclusions**

This study uses a combination of qualitative and quantitative methods to investigate(examine) the effect of both floorplan layout and building size on the efficiency of nursing unit rounds, as well as the effect of both on building energy consumption from the perspective of building energy efficiency. The results of this study show that both floorplan and building size have a significant effect on the efficiency of nursing unit rounds and building energy consumption. By choosing the right floorplan for the number of beds in the ward block, it was possible to increase the efficiency of ward block rounds by 35.68%, save 9.02% of the building's energy consumption and reduce carbon emissions by 23,423 kg. Based on this study, the optimal floorplan types for different numbers of beds in cold regions and the range of the optimal number of beds for different floorplan types are derived, thus providing a reference for achieving a floorplan with high rounding efficiency and low building energy consumption, which has certain practical significance for the planning and architectural design of ward buildings. The sample size and types

used in this thesis study are limited and cannot cover all types of ward-building floorplans, so case studies should be conducted on a case-by-case basis in the design of ward buildings. In addition, factors such as the function of use, number of patients, and length of stay in different types of ward buildings may have an impact on roving efficiency and energy consumption, and these factors need further consideration.

**Author Contributions:** Conceptualization, C.J. and Q.D.; methodology, C.J. and Q.D.; software, C.J.; validation, Q.D. and C.J.; formal analysis, C.J. and Q.D.; investigation, X.S. and C.J., resources, C.J. and X.S.; data curation, C.J., J.Z., and G.W.; writing—original draft preparation, C.J. and Q.D.; writing—review and editing, Q.D., C.J., and G.W.; visualization, C.J. and J.Z.; supervision, Q.D. and G.W.; funding acquisition, Q.D. All authors have read and agreed to the published version of the manuscript.

**Funding:** This research received no external funding.

**Data Availability Statement:** Data derived from the current study can be provided to readers upon request.

**Conflicts of Interest:** The authors declare no conflict of interest.

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
