# Peer review of "A Study on the Layout of Hospital Ward Buildings in Cold Regions of China Based on the Efficiency of Nurse Rounds"

_buildings, doi:10.3390/buildings13061399_

Round 1

Reviewer 1 Report

This manuscript proposes a multi-step framework for evaluating the building shape factor’s impact on hospital building efficiency and energy consumption. The work overall leverages a great number of empirical methods as well as simulation tools. However, for many of the models deployed, parameter selection and validation were not clearly presented. My general feedback would be that there are lots of citations that have issues being properly shown, and some trimming work could be used with greater attention to the details. Another main concern I have is that I didn’t see many results of relationship between the shape factor of layouts and building energy consumption, which should be content that is of the most relevance to such journals/special issues. The current main values I see are regression analysis on the simulation path results with a lack of how energy consumption varies in different scenarios. To my understanding,  varying window-to-wall ratios in different layout could also contribute to the consumption changes. Please consider providing the detailed shape factors for each layout and simulated building energy performance as additional table/results and propose the optimal strategy for each type of layout. I also have the following specific comments and questions :

1. Line 47 has a typo please fix it

2. Has Figure 1 considered the impact of COVID-19? Please add an explanation of how COVID-19 was adjusted for Figure 1 and add a citation in the main text. (Line 33)

3. Please consider updating Figure 2’s region name to bold with a larger font size.

4. Line 210-215, please either add a reference or further elaborate on how these simulation parameters are selected: the inpatient department is set up on 13 floors with a floor height of 4.2m. If these are the most common or standard building parameters for hospitals, please add a reference.

5. Line 238, reference missing.

6. Line 241 to 246, please add a reference for each of the three evaluation metrics

7. Line 271, reference missing.

8, Table 3 could be trimmed to maybe just 3 examples from each type of layout (Min, Max, and Median number of beds)

9, Line 348, Figure number has a typo

10, Please update the R2 either to R2 or R-Squared and Add the P-value for each regression model summary (If the author is naming the P-value as Sig. please change the metric name, as P-value is more internationally acceptable)

Please fix some of the citation issues

Reviewer 2 Report

This study developed a methodology for floor layout optimization for hospital ward buildings using BIM. The topic is interesting and suitable for this journal. However, this manuscript needs to be further improved in its novelty and academic contribution before accepting it for publication. My comments are as follows:

1.  Citation of Autodesk Revit is missing.

2. Please fix “Error! Reference source not found” and “Error! Bookmark not defined” errors.

3. Please explain whether the proposed methodology can be applied when using other BIM authoring software, such as Archicad. If so, please elaborate.

4. Please explain whether the proposed methodology supports openBIM, e.g. the use of IFC. If so, please elaborate.

5. Daylighting and solar heat are not considered and should be explained with the consideration of cold regions.

6. While reducing the energy consumption of building operations decreases operation carbon emissions, it may increase embodied carbon emissions during construction. The evidence shows that it is a low-carbon design is lacking as the part related to embodied carbon is missing.

7. The limitations and possible future works of this study should be presented in an organized manner.

The word "correlation" has been used repeatedly. Please consider using the word "proportional".

Reviewer 3 Report

The article attempts to indicate the optimal functional arrangement of floors in hospitals in the context of the nursing staff's work efficiency and the shape factor as one of the parameters determining the energy efficiency of the building. Appropriate analysis was performed using mainly commonly used software. Nevertheless, I believe that the manuscript lacks some elements characteristic of scientific articles, and there are also some inaccuracies, mainly in the definition of parameters.

In connection with the above, I recommend further processing of this publication in the journal Buildings after taking into account the suggestions described below.

First of all

1. "Equations" 1-7 are written in text, which is unacceptable in scientific articles. The equations are also embedded in the text, making e.g. the paragraph (lines 325-332) difficult to understand. Individual parameters should be marked with symbols described in the text below, as in Equation 8. In addition, there are inaccuracies: e.g. in equations 5 and 7: "Distance from the nursing activity centre to the bed" - which bed? After all, there are many in the ward. If it is an average, write it down and use the sum sign in the formula.

2. The authors mostly omit units, which is disqualifying. This must be made visible in all formulas, tables, etc. For example, in what units is "distance ...": [m]??? [km]???. Also in Equation 8: "building shape factor" - it should be [1/m]. Units are also omitted from Table 6.

3. (1 - average distance from the nurses' station to the ward %) etc.

Parameter designations are completely bizarre. Besides, if this is "distance" then how come it is given as a percentage? How did the authors translate the results from Table 4 into the results shown in Figure 3 (line 346)?

4. Table 2. Column descriptions mixed up.

What does "Number of wards", "Ward Building Total area" etc. mean?

What are the floor areas shown. After all, these values should be different for each of the three functional systems.

5. Shape factor.

What is the detailed methodology for calculating this parameter? Was it calculated for entire buildings or for one floor.

In addition, the authors must clearly write that this parameter is only one of the factors determining the energy efficiency of the building, and in principle it affects the calculated heat losses per unit of volume (assuming equal insulation of external envelopes). It is also obvious that the larger the building is, the more the shape factor decreases.

6. Conclusions should be more specific and a little longer. It is necessary to assess to what extent the aim of the article has been achieved and combine the two researched issues: staff efficiency and shape factor analysis.

The sentence (lines 462-465) should be deleted. Such lofty wording does not raise the substantive level of the article.

Aside from that:

7. What is b2 in table 5? In addition, linear and quadratic regression functions were used. Why are other types of functions omitted?

8. Figure 3 appears in the article 3 times. There are phrases in the text such as "Figure?" "Figure a".

9. Descriptions in Figure 3 (line 164): "Unit2 Single Corridor Layout, Unit3 Double-Corridor Layout (Racetrack)" - probably incorrect.

10. Lines 117-122. I think a reference is needed.

11. Lines 152-153. „Redial” or „Radial”?

12. Line 221. What is „appropriate proportion of nursing unit wards”?

13. Lines 390-391 – “Relationship between the layout and size of nursing unit layout and building energy consumption”. This is not true. Where was the "building energy consumption" calculated here?

14. Line 394 – “Body size factor”. Why? After all, in the article this parameter was called "(building) shape factor". There are also other similar inaccuracies in the text.

15. Need to improve the quality of most graphs in Figures. The descriptions are especially hard to read.

Round 2

Reviewer 1 Report

I reviewed the revised paper, and appreciate that the authors added some further nuance to their text and I like how the discussion section has different layers. One minor suggestion I have is to remove the 'In terms of' in each caption of the discussion.

N/A

Author Response

Thank you again for your comments on our manuscript. According to your comment, 'on' has been deleted.

Reviewer 2 Report

The authors have addressed some of the crucial issues identified in the previous version of the manuscript, though the developed methodology relies on Revit, which is a proprietary BIM software. The manuscript should be considered for publication in the journal.

Author Response

Thank you again for your feedback and valuable suggestions on improving the quality of our manuscript.

Reviewer 3 Report

After analyzing the revised text sent to me, I find that the article has been improved and my comments in the previous review have been sufficiently taken into account.

I thank the authors for responding to my comments.

Minor remarks:

Not corrected (redial -> radial) on line 154.

I still do not understand what "Each nursing unit Number of wards" means in Table 2.

In the headings of Tables 2, 4 and 5, and in Figure 3, there is a discrepancy in the use of lowercase and uppercase letters (some words capitalized, others lowercase). There are also many such inconsistencies in the text. e.g. line 257: "Distance difference", line 345 "Distance Difference" etc.

Section 3.3 "Shape factor and building energy consumption"

Where is the “shape factor” in the content of this section now? Please change the section title.

Lines 390-405. In my opinion, this text should be included in section 3.3

Equation 8: incorrect units: Energy consumption per bed, I think: [kWh/(hb×a)], Total Energy consumption, I think: [kWh/a]. Also: in formula descriptions write units in square brackets.

Table 5: Axis description: “Distance coefficient” – unit omitted.

Improve the quality of some illustrations. For example, the graphs in Tables 5 and 7 are completely illegible. Perhaps they need to be enlarged and arranged "top-down", not "left-to-right".
